# Vaccination in Pregnancy against Pertussis: A Consensus Statement on Behalf of the Global Pertussis Initiative

**DOI:** 10.3390/vaccines10121990

**Published:** 2022-11-23

**Authors:** Bahaa Abu-Raya, Kevin Forsyth, Scott A. Halperin, Kirsten Maertens, Christine E. Jones, Ulrich Heininger, Daniela Hozbor, Carl Heinz Wirsing von König, Amar J. Chitkara, Rudzani Muloiwa, Tina Q. Tan

**Affiliations:** 1BC Children’s Hospital Research Institute, Vancouver, BC V5Z 4H4, Canada; 2Department of Pediatrics, University of British Columbia, Vancouver, BC V6H 3V4, Canada; 3College of Medicine and Public Health, Flinders University, Adelaide, SA 5042, Australia; 4Canadian Center for Vaccinology, Departments of Pediatrics and Microbiology and Immunology, Dalhousie University, Izaak Walton Killam Health Centre, and Nova Scotia Health Authority, Halifax, NS B3H 4R2, Canada; 5Centre for the Evaluation of Vaccination, Vaccine and Infectious Diseases Institute, University of Antwerp, 2610 Antwerp, Belgium; 6Faculty of Medicine and Institute for Life Sciences, University of Southampton, Southampton SO17 1BJ, UK; 7NIHR Southampton Clinical Research Facility and NIHR Southampton Biomedical Research Centre, University Hospital Southampton NHS Foundation Trust, Southampton SO16 6YD, UK; 8Pediatric Infectious Diseases, University of Basel Children’s Hospital, 4056 Basel, Switzerland; 9Instituto de Biotecnología y Biología Molecular, Facultad de Ciencias Exactas, Universidad Nacional de La Plata, CONICET, La Plata 1900, Argentina; 10Independent Researcher, 47805 Krefeld, Germany; 11Department of Pediatrics, Max Super Speciality Hospital, New Delhi 110017, India; 12Department of Paediatrics and Child health, Faculty of Health Sciences, University of Cape Town, Cape Town 7935, South Africa; 13Feinberg School of Medicine, Northwestern University, Chicago, IL 60208, USA

**Keywords:** pertussis, pregnancy, vaccination, immunity, protection, Global Pertussis Initiative

## Abstract

**Highlights:**

**Abstract:**

Infants are at high risk for severe morbidity and mortality from pertussis disease during early infancy. Vaccination against pertussis in pregnancy has emerged as the ideal strategy to protect infants during these early, vulnerable, first months of life. On 30 November and 1 December 2021, the Global Pertussis Initiative held a meeting that aimed to discuss and review the most up-to-date scientific literature supporting vaccination against pertussis in pregnancy and outstanding scientific questions. Herein, we review the current and historically published literature and summarize the findings as consensus statements on vaccination against pertussis in pregnancy on behalf of the Global Pertussis Initiative.

## 1. Introduction

Pertussis, or “Whooping cough”, is a respiratory infection caused by *Bordetella pertussis* [1]. Despite the availability of both whole-cell pertussis (wP) and acellular pertussis (aP) vaccines, resurgence of *B. pertussis* was observed in a number of countries in the first decade of the 21st century, with young infants disproportionally affected by severe morbidity and mortality [2,3,4,5,6]. In order to close the gap of high susceptibility of newborns and young infants to pertussis between birth and the completion of the primary series of immunizations, an increasing number of countries have recommended vaccination against pertussis for pregnant women since 2011 [7].

Vaccination against pertussis in pregnancy induces *B. pertussis*-specific immune responses in the mother and the transfer of anti-*B. pertussis* antibodies across the placenta and via the breast milk to protect the infant from pertussis during early infancy. A large body of evidence supports that vaccination against pertussis in pregnancy is safe for pregnant women, the developing fetuses and infants [8,9,10,11,12,13,14,15,16,17,18,19,20]. The protective effects of vaccination against pertussis in pregnancy were evident by the decrease in the incidence of severe pertussis in young infants shortly after implementation of this strategy [21,22,23,24,25]. In real-world studies, vaccine effectiveness (VE) ranged from 69 to 93%, 91 to 94% and 95% for prevention of laboratory-confirmed infection, hospitalization and mortality due to pertussis in infants <2–3 months in high-income countries (e.g., US, England, Spain, Australia) [8,21,22,26,27,28,29,30]. Vaccination against pertussis in pregnancy was also shown to be protective in studies from middle-income countries [31]. In Brazil, this strategy was 82% protective in the prevention of laboratory-confirmed pertussis infection in infants <2 months of age [23], and prevention of all pertussis infant deaths in Argentina in the year following introduction of pertussis vaccination in pregnancy was achieved [32].

On 30 November and 1 December 2021, the GPI held a virtual expert meeting with 30 pertussis experts to discuss vaccination in pregnancy, with representation from 18 countries from around the globe. The objectives of the meeting were to: (1) review the currently available scientific data supporting vaccination in pregnancy (ViP) to protect young infants from pertussis disease, hospitalization and death; (2) review the outstanding scientific questions related to pertussis ViP; (3) discuss global ViP recommendations, as well as the status of vaccine coverage and implementation across the world.


**About the Global Pertussis Initiative (GPI)**
The GPI was initiated in 2001 to raise global awareness about pertussis, develop evidence-based recommendations for vaccination strategies to reduce the disease burden in infants and prevent the waning of immunity in older children and adolescents. To achieve these goals, the GPI convenes global and regional meetings, attended by experts from specialized fields, who work together to achieve a consensus on recommendations for immunization strategies that will be acceptable at local, national and regional levels.

This paper summarizes the scientific data presented at the meeting, the discussion and the outstanding questions in the field of pertussis ViP and was subsequently supplemented with a literature review.

## 2. Immunobiology of Pertussis Vaccination in Pregnancy

The maternal immune system undergoes dynamic changes during gestation to tolerate the semi-allogeneic fetus while still preventing infections in the pregnant woman and the fetus. Despite the normal immune modulation that occurs during pregnancy, the antibody- and cellular-mediated immunity of aP vaccines in pregnant and non-pregnant women have not been found to be significantly different [33]. Recently, using the systems vaccinology approach, vaccination with an aP vaccine was shown to induce upregulation in the type I interferon response and innate immunity genes in both pregnant and non-pregnant hosts, also suggesting that pregnancy does not affect vaccine-induced immune responses [34].

Immunoglobulin G (IgG) is the main Ig isotype that crosses the placenta to the infant. IgG transfer begins towards the end of the first trimester of pregnancy and increases as pregnancy advances, reaching concentrations that are higher in the infant than in the mother at term delivery [35,36,37,38]. Active trans-placental transfer of IgG into the fetal circulation is mediated by neonatal Fc receptors (FcRn), localized in the syncytiotrophoblast of the placenta [39]. The structure and function of IgG affect the efficiency of the transfer of IgG across the placenta. FcRn binds to the CH3 domain of all IgG subclasses [40], however, differences in IgG subclass transfer have been detected [40,41], including enhanced binding to IgG1 and differential transfer efficiencies of allotypic variants of IgG3 known to bind FcRn with different affinities [42]. Analysis of 17 studies reported that the efficiency of the mean transfer of the different IgG subclasses across the placenta followed an IgG1 > IgG3 = IgG4 > IgG2 hierarchy [43]. This is important as pertussis vaccines include proteins as target vaccine antigens and, thus, vaccination with pertussis vaccines is expected to induce IgG1 and IgG3 subclasses [44,45], which are expected to be efficiently transferred across the placenta. Indeed, the efficiency of the transfer of different anti-*B. pertussis* antigen-specific IgG1s (pertussis toxin (PT), filamentous hemagglutinin (FHA), pertactin (PRN), fimbriae (FIM)) across the placenta was found to be the highest compared with IgG2, 3 and 4, among a mixed cohort of women vaccinated and unvaccinated against pertussis in pregnancy [46]. The efficiency of the transfer of anti-*B. pertussis* IgG with different functions also varies. IgG antibodies mediating phagocytosis had variable efficiencies of trans-placental transfer, while IgG antibodies mediating natural killer (NK) cell activation were consistently higher in cord than in maternal sera for PT, FHA, PRN and FIM, indicating that these antibodies are preferably transferred across the placenta [46]. Transfer of maternal antibodies is also affected by maternal health status, with infection with human immunodeficiency virus (HIV) associated with a lower transfer of vaccine-specific antibodies and alterations of biophysical features [47,48,49,50]. The immune response of pertussis vaccines in women living with HIV has not been largely explored.

Immune responses against pertussis vaccination peak and wane rapidly after vaccination [51]. Following vaccination against pertussis in pregnancy with different aP formulations, anti-*B. pertussis* antibodies against all *B. pertussis* antigens increased one month after vaccination, followed by a rapid decline within the first year after delivery [17,33,52,53,54]. Therefore, to provide optimal protection to the infants, vaccination has been recommended for every pregnancy regardless of the interval between subsequent pregnancies. Of the antibodies against different *B. pertussis* antigens, those against PT are probably most important as they prevent severe disease due to leukocytosis in the young infant when infected with *B. pertussis* [55].

## 3. Optimal Timing of Vaccination

The optimal timing of vaccination against pertussis in pregnancy is the time window in pregnancy where administration of a pertussis vaccine is associated with the highest VE in young infants, the highest transfer of *B. pertussis*-specific antibodies to the fetus and the least inhibition of the infants’ immune responses to their own vaccines administered in infancy [56].

### 3.1. Time-Dependent VE

Studies have attempted to address the effect of timing of vaccination against pertussis in pregnancy on VE and results have been inconclusive. Observational studies from the US showed that vaccination in the third trimester (27–36 weeks’ gestation (WG), the vaccination timeframe recommended by the US Centers for Disease Control and prevention) was more protective in the prevention of laboratory-confirmed pertussis in infants <8 and <12 weeks old and born at term when compared with vaccination outside 27–36 WG [28]. In addition, a study from the US reported that infants <18 months of age whose mothers received the pertussis vaccine <27 WG did not have reductions in pertussis rates compared with infants of unvaccinated mothers, although this study is limited by the small number of cases leading to a wide confidence interval (CI) (hazard ratio (HR) for pertussis = 1.10, 95% CI = 0.54–2.25) [57].

A further study from the US did not reach a definite conclusion whether vaccination early in the third trimester (27–31 WG) was associated with higher protection than vaccination later in the third trimester (32–36 WG), likely due to the small number of cases [28]. A study from Spain reported that the VE of vaccination against pertussis in pregnancy was 88% (95% CI = 53.8%–96.5%) in the prevention of laboratory-confirmed pertussis in infants <2 months of age, and the VE was not different in infants of mothers vaccinated during 26–31 WG compared with ≥32 WG [58]. However, this study was not powered to detect time-dependent difference in VE.

Prematurity is a significant and independent risk factor for severe *B. pertussis* morbidity and mortality [5,59]; therefore, vaccination against pertussis in pregnancy also needs to provide protection for this vulnerable group. In the US, vaccination against pertussis in pregnancy (most women received aP vaccine during 27–36 WG) was associated with a VE of 91% (95% CI: 64–97%) in preventing pertussis in preterm infants <6 months of age [60]. However, the exact timing of vaccination in pregnancy and gestational age at birth of preterm infants were not clearly described. Extending vaccination against pertussis in pregnancy to early gestation is expected to be associated with higher protection of preterm infants from pertussis who may miss the protection if born earlier than, or a short time after, aP administration in pregnancy. Following the extension of the timing of vaccination against pertussis in pregnancy from 28–32 WG to 20–32 WG in the UK, a decrease was seen in the number of preterm infants <60 days of age hospitalized with pertussis during a 19-month period of the study, from 20 to 9 cases, while cases of term infants remained the same (62 and 60 cases before and after the change, respectively) [61]. However, this decrease in pertussis cases in preterm infants was nonsignificant and was associated with an increase in vaccine uptake from 60% to 70% during the study period and the number of preterm infants with pertussis was low prior to extending the timing of vaccination. These could have contributed to the decrease in numbers reported, making definite conclusions on the sole effect of timing of vaccination in pregnancy difficult [61,62].

Whilst the optimal timing of pertussis vaccination in pregnancy is still debated, it is well established that being vaccinated in pregnancy provides significant protection to the infant and thus vaccination any time in pregnancy should still be encouraged [63,64].

### 3.2. Time-Dependent Immunogenicity

As a result of the inconclusive VE data (as detailed in the preceding section), immunogenicity studies were important in guiding the recommendations of optimal timing of vaccination against pertussis in pregnancy in different countries, assuming that higher concentrations of anti-*B. pertussis* antibodies, especially those against PT, are associated with higher clinical protection from clinical disease [65].

One study from Switzerland showed that cord anti-PT and anti-FHA IgG concentrations were higher in women vaccinated between 13–25 WG compared with those vaccinated after 25 WG gestation in both term [66] and preterm infants [67]. Another study from Argentina did not find significant differences between cord anti-PT IgG concentrations following second (13–26 WG) vs. third (27–36 WG) trimester vaccination [68]. However, the number of subjects in the latter study was rather small, leading to very broad confidence intervals (CI) of antibody levels indicating that the study was underpowered to demonstrate the effect of timing of vaccination in pregnancy.

Several studies have shown that early third trimester vaccination against pertussis is associated with statistically significant higher anti-*B. pertussis* antibody concentrations and avidity than vaccination in the late third trimester of pregnancy [69,70,71,72,73,74,75]. In contrast, other studies did not report statistically significant differences in concentrations or avidity of cord sera antibodies against PT following early third compared with late third trimester vaccination [76,77], although a trend was observed for higher anti-PT levels for vaccination earlier vs. later in pregnancy [78].

### 3.3. Time-Dependent Effect on Infant Immune Response

A recent meta-analysis showed that there was no association between the timing of the tetanus–diphtheria–acellular pertussis (Tdap) vaccination in pregnancy and the infants’ subsequent immune response [78]. Specifically, timing of administration of Tdap in pregnancy did not affect post-primary and post-booster immunization anti-*B. pertussis*, anti-tetanus-toxoid (TT) and anti-diphtheria-toxoid (DT) IgG levels [78].

## 4. Modification of Infant Immune Response to Vaccination following Vaccination against Pertussis in Pregnancy

Vaccination against pertussis in pregnancy is associated with modification of an infant’s active immune responses to their own vaccine antigens [79,80,81]. This has largely led to lower anti-*B. pertussis* antibody concentrations after primary and booster vaccination series in infants born to vaccinated mothers compared with infants born to mothers who are unvaccinated. Thus, the phenomenon has largely been referred to as “interference” or “blunting” [82,83]. There is still excellent boosting in this situation; however, the boosting is somewhat less than in the non-pregnancy vaccinated infants. Given that currently available pertussis vaccines are combined with TT and diphtheria-toxoid DT in the form of Tdap vaccines, receipt of Tdap in pregnancy has also been associated with higher antibody concentrations against TT and vaccines conjugated to TT, such as *Haemophilus influenzae b* (Hib) pre- and post-infant vaccination. Given that vaccination in pregnancy with this combination vaccine may have different effects on different vaccine responses, the phenomenon is not accurately described by “interference” or “blunting”, leading to a call for a more neutral, accurate and broad term of “modification” or “modulation” of immune responses following vaccination in pregnancy [84]. In this paper, the term “modification” is used to describe the overall effect of vaccination in pregnancy on overall antibody responses in infants, and the term “interference” describes a specific decrease in antigen-specific antibody responses in infants of vaccinated mothers.

### 4.1. Effect on Immune Responses to B. pertussis Antigens in aP Vaccines

Vaccination against pertussis in pregnancy is associated with significantly lower anti-*B. pertussis* IgG concentrations in infants born to vaccinated compared with unvaccinated women after their primary and booster vaccination [17,82,85,86,87,88,89,90]. Individual participant data meta-analysis of 10 studies (9 performed in high-income countries, 1 performed in a middle-income country) has recently shown lower anti-*B. pertussis* IgG concentrations after primary and booster vaccination for pertussis antigens included in aP vaccines [91]. No study, to our knowledge, was performed in a low-income country.

#### 4.1.1. Preterm Infants

Preterm infants are not only at risk for pertussis in early infancy but may also be more vulnerable to consequences of lower anti-*B. pertussis* antibody concentrations post-vaccination, which may put them at risk for pertussis disease later in their infancy [92]. A study from the UK showed that anti-FHA concentrations were significantly lower in preterm infants born to vaccinated compared with unvaccinated mothers after primary vaccination, but this resolved by 12 months of age [93]. Recently, equal anti-*B. pertussis* antibody concentrations were achieved after primary vaccination in term and preterm infants born to women vaccinated against pertussis in pregnancy. After booster vaccination, significantly lower anti-PT and anti-FHA IgG concentrations were reported in preterm compared with term infants of vaccinated women, but these concentrations were comparable to those achieved in both preterm and term infants of unvaccinated women [94].

#### 4.1.2. Clinical Relevance in Countries Using aP for Infant Vaccination

While immunogenicity data could suggest that infants of women vaccinated against pertussis in pregnancy might be at increased risk for pertussis later in childhood, surveillance data from the UK have not demonstrated an increase in pertussis cases in toddlers (1–4 years of age) after the introduction of the vaccination in pregnancy program [21,95]. A large study from the US also reported that between 6 and 18 months of life there were no differences in pertussis rates by receipt of antenatal aP vaccine and after adjustment for the infant’s diphtheria, tetanus and acellular pertussis (DTaP) receipt (HR = 0.69, 95% CI = 0.26–1.86 for pertussis and HR = 2.60, 95% CI = 0.15–46.2 for inpatient-only pertussis), further suggesting that there is no clinical significance of reduced anti-*B. pertussis* antibody concentrations in infants of vaccinated mothers [57]. Another study from the US reported the benefit of vaccination against pertussis in pregnancy and showed a VE of 66% (95% CI = 5–88%) in infants at 12 months of age (at which time they had received three doses of DTaP vaccine) [22].

### 4.2. Effect on Immune Response to B. pertussis Antigens in wP Vaccines

In a cohort of US infants born to women unvaccinated against pertussis in pregnancy in the 1990s, vaccination with wP in infants born to mothers with higher anti-PT IgG concentrations was associated with a reduction in infant antibody responses to PT [96], confirming previous findings in infants in Germany [97]. However, this association was not observed in another study after wP vaccination in Pakistan [98]. The magnitude of the decrease in immune responses to *B. pertussis* antigens included in wP infant vaccines following vaccination against pertussis in pregnancy might be different than to aP vaccines, although scant literature has addressed this question. This is important as wP vaccines are used almost exclusively in low- and in middle-income countries for infant vaccination [99]. Lower anti-*B. pertussis* antibody concentrations in infants of vaccinated mothers were observed after wP primary vaccination series when compared with infants primed with aP vaccine and born to mothers vaccinated with aP in pregnancy [100]. It is important to note, however, that antibody functionality was higher in the wP groups [100]. Undoubtedly, more studies are needed to enable definite conclusions. This is of particular importance as wP vaccines in different areas of the world may differ in the quality and quantity of antigens and, thus, the antibody concentrations these vaccines induce.

#### Clinical Relevance in Countries Using wP for Infant Vaccination

Demonstration of the clinical relevance of lower anti-*B. pertussis* antibody concentrations following wP vaccination in infants is more challenging in low-middle-income countries compared with high-income countries due to the lack of comprehensive surveillance systems in some low-income countries, although current data are reassuring [101]. Modeling data from Argentina did not predict clinical significance of reduced anti-*B. pertussis* antibody concentrations and concluded that vaccination in pregnancy may benefit infants aged 2–12 months [102]. These modeling results were later supported by real-world data. In Brazil, the average incidence of pertussis per year for infants aged 1–12 months of age was 64.9/100,000 during the years 2011–2013 (prior to introduction of the vaccination in pregnancy program) and decreased to 29.3/100,000 during 2015–2017 (after the vaccination in pregnancy program was introduced, giving a reduction rate of 0.45 (95% CI: 0.29–0.69)). These data suggest that there is no clinical significance of lower anti-*B. pertussis* concentrations after wP vaccination in infants born to women vaccinated in pregnancy [103]. Recently, Carrasquilla et al. reported on pertussis incidence and mortality in infants <12 months of age who received primary vaccination with wP vaccine before and after introduction of vaccination against pertussis in pregnancy in Colombia [104]. They showed that, following the introduction of vaccination against pertussis in pregnancy, the incidence of pertussis declined by 54.4%, 73.4% and 100% among infants <6, 7–28 and 28–52 weeks of age, respectively. Furthermore, a 100% reduction in pertussis mortality was observed in infants <12 months [104]. These data are critical and reassuring as the study design and age cut-offs chosen (28–52 weeks of age) allowed for the investigation of the clinical significance of lower anti-*B. pertussis* antibody concentrations following wP vaccination in settings of vaccination against pertussis in pregnancy.

Given that data from low-income countries may be challenging to obtain, data from middle-income countries with *B. pertussis* surveillance systems could inform low-income countries lacking such infrastructure. The abovementioned data are also supportive of the recent recommendation by The Technical Advisory Group of the Pan American Health Organization for vaccination against pertussis in each pregnancy in Latin America, where wP vaccines are routinely given to infants [105]. The data also support the WHO recommendation (2015) recommending that national programs administering wP vaccines should continue to use these formulations for primary vaccination [106]. This statement was made before the widespread implementation of vaccination against pertussis in pregnancy programs.

Although data are reassuring, they should still be interpreted with caution and continued surveillance for the clinical impact of lower anti-*B. pertussis* antibody concentrations in infants following vaccination against pertussis in pregnancy in low-middle-income countries should continue. Clinical significance of interference might be evident if the strategy of vaccination against pertussis in pregnancy becomes more widely implemented in low-middle-income countries’ settings, as a larger cohort of infants born to pertussis-vaccinated women will receive wP vaccines.

### 4.3. Modification to DT and TT Antigens and Conjugated Vaccines

Current vaccines against pertussis administered in pregnancy contain DT and TT antigens in addition to the pertussis antigens, which may also modify infant immune response to DT- and TT-containing vaccines. Some vaccines administered in infancy are conjugated to DT and TT as a carrier protein. Thus, it is important to assess the immune responses to these vaccines in infants of vaccinated women.

#### 4.3.1. Effect on Immune Response to DT-Containing Vaccines

While some individual studies reported significantly lower anti-DT antibody concentrations in infants born to women vaccinated against pertussis in pregnancy when compared to infants born to unvaccinated women, other studies did not report this effect, which could be due to lack of power to detect such differences [17,82,85,86,87,88,89]. However, a recent meta-analysis reported significantly lower anti-DT IgG concentrations in infants born to women vaccinated in pregnancy compared with unvaccinated women after the primary vaccine series, before and after booster vaccination with DT-containing vaccines in infancy [91]. It was also shown that infants of vaccinated women had lower anti-*Streptococcus pneumoniae* concentrations for 12/13 serotypes after primary vaccination with 13-valent pneumococcal vaccine conjugated to a non-toxic diphtheria toxin mutant CRM197 (PCV-13) and lower seroprotection rates against 5/13 serotypes [91]. This inhibition of immune response is thought to be mediated by maternally derived anti-DT antibodies transferred to the infants. The clinical significance of these lower concentrations of anti-*Streptococcus pneumoniae* antibodies is unclear and should be assessed by future surveillance studies.

#### 4.3.2. Effect on Immune Response to TT-Containing Vaccines

Individual study results were inconsistent and showed variable results of lower, equal and higher anti-TT concentrations after vaccination in infants born to women vaccinated against pertussis in pregnancy compared with infants from unvaccinated women [17,85,86,87,89]. A recent meta-analysis reported higher seroprotection rates before booster vaccination and higher anti-TT IgG concentrations after booster vaccination in infants born to women vaccinated in pregnancy compared with unvaccinated women, suggesting that vaccination in pregnancy may lead to enhancement of antibody response to the TT component in the infants [91].

Seroprotection rates against *Hib* following vaccination with *Hib* vaccines that are conjugated to TT were higher in infants born to women vaccinated with Tdap in pregnancy when compared with infants of unvaccinated women, providing further support for enhancement to the TT component [91].

### 4.4. Mechanistic Insights into Modification of Immune Responses following Vaccination in Pregnancy

The mechanisms of the association between vaccination against pertussis in pregnancy and modification of immune responses in infants are unclear. Recently, a meta-analysis of eight studies found that higher anti-*B. pertussis* IgG concentrations before primary vaccination in infants of women vaccinated against pertussis in pregnancy were independently associated with lower IgG concentrations to the same *B. pertussis* antigens after primary and booster vaccination [78]. These results suggest that maternal antibodies are an important factor affecting post-natal vaccine-induced immune responses; however, the cellular mechanisms leading to these decreased antibody responses are not known and different hypotheses have been proposed. Inhibition of B-cell responses to vaccine antigens administered to infants by maternal antibodies through both epitope masking and neutralization of vaccine antigens has been proposed as a possible mechanism [83,107,108]. However, these mechanisms do not explain the long-term effect on the infant immune response measured after booster vaccination, a time period when maternal antibodies are not expected to circulate in infant blood. Crosslinking of FcγRIIB to the B-cell receptor on B cells via vaccine antigen–antibody complexes leading to inhibitory signals and inhibition of B-cell activation has also been proposed as another possible mechanism [109]. In the mouse model, maternal antibodies against influenza were associated with low number of B cells that differentiate into plasma cells and memory B cells [110]. If these results apply to human infants and pertussis antigens, this could explain the long-term effect of vaccination in pregnancy after maternal antibodies have been removed from infant circulation. Finally, emerging data support that infant T-cell responses to pertussis vaccination are still detected following vaccination against pertussis in pregnancy. T-cell immune responses to PT were detected in infants born to women vaccinated against pertussis in pregnancy after primary and booster vaccination in preterm and term infants [111].

## 5. In Utero Exposure to Vaccine Antigens

While the trans-placental transfer of maternal antibodies following vaccination in pregnancy has been well studied, there are much less data on the effect of vaccination against pertussis in pregnancy on the developing fetal and neonatal immune system [112]. Priming of the fetal immune system to *B. pertussis* antigens after vaccination in pregnancy is important to investigate given that influenza vaccination in pregnancy leads to fetal production of influenza-specific IgM antibodies and influenza-specific CD4^+^ T cells detected in the cord [113]. Recently, T-cell responses to PT were detected in infants born to vaccinated mothers prior to primary vaccination [111]. These results supported an earlier study that showed that T cells from cord samples of newborns born to vaccinated women were activated after stimulation with *B. pertussis* (measured by CD40 ligand and CD69 expression) [114].

Furthermore, infants of vaccinated women were shown to have elevated interleukin (IL)-2 and IL-12 responses, elevated proportions of classical monocyte and reduced monocyte and NK cell cytokine responses at birth compared with infants of unvaccinated women [115]. Taken together, these studies, which suggested modification of adaptive and innate cellular neonatal immune response to *B. pertussis* antigens prior to receipt of first aP vaccine dose in infancy, may support that in utero exposure to pertussis vaccine antigens could happen. However, formal studies confirming these findings are needed and results need to be compared to infants of unvaccinated mothers in larger cohorts to derive definite conclusions.

## 6. Pertussis-Specific Antibodies in Breast Milk following Vaccination in Pregnancy

Studies have shown that vaccination against pertussis in pregnancy leads to induction of *B. pertussis*-specific antibodies in breast milk. Anti-*B. pertussis* secretory immunoglobulin (Ig)A and IgG antibodies were detected in colostrum and breast milk of women vaccinated against pertussis in pregnancy during the first 8 weeks after term delivery [116,117]. Recently, it was reported that anti-PT IgA and IgG concentrations in the colostrum were comparable in breast milk after term and preterm delivery and that anti-PT antibodies were detected up to 12 weeks after delivery [118]. The added benefit of these antibodies in breast milk to the clinical protection provided via trans-placental transfer of maternal antibodies has not been studied. In this regard, an important question is how anti-*B. pertussis* antibodies in breast milk may provide clinical protection to the infants. It is possible that natural regurgitation of breast milk during and after breastfeeding could contribute to mucosal immunity in the neonatal upper respiratory tract. Another possibility is that *B. pertussis*-specific antibodies in breast milk can be absorbed across the intestine into the neonatal blood circulation. *B. pertussis*-specific antibodies from mother’s breast milk were stable in the gastrointestinal tract during digestion of preterm infants, suggesting that absorption of anti-*B. pertussis* antibodies across the intestine may be possible [119].

## 7. Effect of COVID-19 on Pertussis Detection and Immunity

There have been multiple reports of a significant decrease in detection of *B. pertussis* infections at the population concentrations associated with COVID-19 mitigation measures [120,121,122,123]. Data from United Kingdom Health Security Agency have recently shown a significant decrease in the incidence of pertussis in the pandemic years 2020 and 2021 across all age groups (<1, 1–4, 5–14 and >15 year(s) of age) compared with pre-pandemic years [124]. Data from the province of Ontario, Canada showed that only 3 cases of pertussis were reported between January and June 2021, a profoundly lower reported case load compared with 193 cases as the 5-year average year-to-date count [125]. In British Columbia, Canada, the incidence of confirmed *B. pertussis* declined in the years 2020 and 2021 compared to pre-pandemic years in all age groups [126]. In this context, recent data have shown that anti-*B. pertussis* antibody concentrations decreased significantly in women of childbearing age in the first year of the COVID-19 pandemic, in the context of mitigation measures and in the absence of recent boosting with pertussis vaccines, indicating that *B. pertussis* pre-existing immunity declines in the context of the absence of circulation of *B. pertussis* [126]. Altogether, these data are important as they suggest that enhanced surveillance of *B. pertussis* needs to continue in the era of relaxing measures to control COVID-19 and should be coupled with the continuous recommendation of the need of vaccination in pregnancy so as to protect young vulnerable infants from severe pertussis disease during infancy. This is vital as reduced immunization coverage against *B. pertussis* has been reported during the COVID-19 pandemic [127].

## 8. Concluding Remarks

Following group discussions during the virtual GPI expert meeting, held on 30 November and 1 December 2021, and a review of the literature on vaccination against pertussis in pregnancy, the following consensus statements on vaccination against pertussis in pregnancy were made on behalf of the GPI (Table 1).

## Figures and Tables

**Table 1 vaccines-10-01990-t001:** Global Pertussis Initiative Statement on Vaccination Against Pertussis in Pregnancy.

Safety and Vaccine Effectiveness
Vaccination against pertussis in pregnancy is safe for pregnant women and newborns and highly effective in preventing pertussis in young term and preterm infants.
**Timing of vaccination**
Vaccination against pertussis in the third trimester of pregnancy is highly effective in the prevention of pertussis in both term and preterm infants.
A growing body of evidence supports that vaccination early in the third trimester is associated with higher newborn anti-*B. pertussis* antibody concentrations compared with vaccination in the late third trimester. More data are required to confirm this observation and whether vaccination early in the third trimester is associated with higher vaccine effectiveness compared to later in the third trimester.
More studies are needed to determine whether vaccination in the second trimester is associated with higher newborn anti-*B. pertussis* antibody concentrations and vaccine effectiveness compared with vaccination in third trimester of pregnancy.
There is currently no evidence to suggest that timing of vaccination in pregnancy affects post-primary and post-booster vaccination antibody concentrations in infants, although more formal studies designed to answer this question are needed.
Vaccination against pertussis at any time during pregnancy is still important to ensure protection of infants against pertussis.
**Modification of an infant’s immune responses**
The term “modification” or “modulation” describes the overall effect of vaccination in pregnancy on overall antibody responses in infants, and the term “interference” describes a specific decrease in antigen-specific antibody responses in infants of vaccinated mothers.
A significant body of literature supports that (A) infants of vaccinated women have less boosting of anti-*B pertussis* antibody concentrations after their own vaccination and (B) this is not clinically significant in countries using aP vaccines for primary and booster vaccination. More immunogenicity and vaccine effectiveness studies are needed in countries using wP vaccines, although current literature does not show this to be of clinical relevance (at this time).
Infants of vaccinated women have lower anti-*Streptococcus pneumoniae* IgG concentrations after vaccination with pneumococcal vaccines conjugated with CRM197, but the clinical significance needs to be investigated. Formal studies to answer this question are needed.
Infants of vaccinated women have higher anti-TT concentrations compared with infants of unvaccinated women, which is not adequately described by “interference” or “blunting”.
The mechanism of modification of immune responses in infants needs to be investigated, although it is likely to be mediated, at least in part, by high maternal antibody concentrations.
**In utero exposure to vaccine antigens**
Recent literature supports that there might be an exposure to vaccine antigens in utero as evident by *B. pertussis*–specific cellular immune response in infants of vaccinated women prior to receipt of any of the infant’s vaccine doses. The clinical significance of this is unclear.
Non-specific effects following vaccination against pertussis in pregnancy have not been studied and should be addressed by future research.
**Anti-*B. pertussis* antibodies in breast milk**
Vaccination in pregnancy induces anti-*B. pertussis* antibodies in breast milk until 12 weeks post-partum. The added benefit of breastfeeding in infants of vaccinated women to clinical protection is unclear.
**Effect of COVID-19 on pertussis**
COVID-19 mitigation strategies have resulted in a significant decrease in *B. pertussis* circulation, which could affect population immunity against *B. pertussis*. Continued enhanced surveillance during COVID-19 and emphasis on continued vaccination is needed.

## Data Availability

Not applicable.

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
