# Peer review of "Vaccination in Pregnancy against Pertussis: A Consensus Statement on Behalf of the Global Pertussis Initiative"

_vaccines, 2022, doi:10.3390/vaccines10121990_

Round 1
Reviewer 1 Report
The present article represents the extended report of a consensus conference that was held as a virtual meeting on November 30th and December 1st, 2021. Authors have further enhanced the original content by an up-to-date review of newly published studies: even though the present study could be defined a NARRATIVE review, Authors have performed a decent and honest work in including also studies that may be quite uncomfortable in promoting maternal vaccination (see section 4.1).
The content is well organized, and Table 1 summarizes accurately the meaning of the main text, and mostly stresses the importance of improving the surveillance in the post-COVID-19 era (e.g. Continued enhanced surveillance during COVID19 and emphasis on continued vaccination is needed).
I've no specific recommendations or requirements and, from my point of view, the present study could be accepted as it is.
Reviewer 2 Report
Well done on a well written and comprehensive treatment of the topic. A few minor points. It is becoming apparent that the gold standard for vaccine effectiveness is a reduction in all-cause mortality, i.e., vaccines have both specific and non-specif effects. Are there any studies which address this point and perhaps highlight as a research gap.
The involvement of small RNAs (miRNAs) in infection and vaccination is increasing in awareness. Any studies to date? If not highlight this as a research gap.
This is a well written paper by a pannel of international experts on this topic. As such the list of references are comprehensive and well reviewed.
